# Semantic Bimodal Presentation Differentially Slows Working Memory Retrieval

**DOI:** 10.3390/brainsci13050811

**Published:** 2023-05-17

**Authors:** Jia Cheng, Jingjing Li, Aijun Wang, Ming Zhang

**Affiliations:** 1Department of Psychology, Research Center for Psychology and Behavioral Sciences, Soochow University, Suzhou 215123, China; 2Department of Psychology, Suzhou University of Science and Technology, Suzhou 215009, China; 3Faculty of Interdisciplinary Science and Engineering in Health Systems, Okayama University, Okayama 700-0082, Japan

**Keywords:** multisensory congruency, bimodal n-back, concrete words, abstract words, working memory, modal representations

## Abstract

Although evidence has shown that working memory (WM) can be differentially affected by the multisensory congruency of different visual and auditory stimuli, it remains unclear whether different multisensory congruency about concrete and abstract words could impact further WM retrieval. By manipulating the attention focus toward different matching conditions of visual and auditory word characteristics in a 2-back paradigm, the present study revealed that for the characteristically incongruent condition under the auditory retrieval condition, the response to abstract words was faster than that to concrete words, indicating that auditory abstract words are not affected by visual representation, while auditory concrete words are. Alternatively, for concrete words under the visual retrieval condition, WM retrieval was faster in the characteristically incongruent condition than in the characteristically congruent condition, indicating that visual representation formed by auditory concrete words may interfere with WM retrieval of visual concrete words. The present findings demonstrated that concrete words in multisensory conditions may be too aggressively encoded with other visual representations, which would inadvertently slow WM retrieval. However, abstract words seem to suppress interference better, showing better WM performance than concrete words in the multisensory condition.

## 1. Introduction

Working memory (WM) is typically considered a system with limited capacity that can process, store, and monitor information in a short period [1], which is the ability and/or resources to concurrently process and store task-relevant information or coordinate the processing of multiple competing inputs [2]. Some studies have reported that WM involves the temporal maintenance of an active representation of external perception information so that it is available for subsequent retrieval processing [3]. Recent evidence suggests that brains tend to aggregate and analyze information from different sources with connections to form a unified cognitive representation [4] and demonstrate a bimodal WM retrieval advantage [5]. Thus, multisensory integration may be necessary to form a multisensory memory representation [6]. Notably, some evidence has revealed that the congruency relationship in multisensory stimuli seems to play an important role in the formation of multisensory memory representations [7,8].

The multisensory congruency relationship reflects the tendency to associate seemingly arbitrary features or stimulus dimensions that have a certain relationship under multisensory conditions [3,7]. Previous multisensory studies have indicated that multisensory integration efficiency was modulated by high-level semantic relationships between different sensory modalities [9]. Recently, many studies found that not only enhanced perceptual behavioral performance but also accelerated WM retrieval was observed when visual and auditory stimuli shared common rather than conflicting semantic information [3,8]. For example, Xie et al. (2017) reported faster visual WM retrieval in a semantically congruent audiovisual WM encoding condition [10]. Further standardized low-resolution brain electromagnetic tomography (sLORETA) results revealed that the posterior parietal cortex (PPC) could play a central executive role in integrating sensory information from the visual–spatial sketchpad and phonological loop into a unified multisensory representation, resulting in faster WM retrieval. Moreover, the congruency relationship in multisensory stimuli at other levels in addition to the semantic level has been explored [5,7]. For example, Brunetti et al. (2017) found that three different types of cross-modal congruency relationships (audiovisual numerosity, pitch/elevation, and pitch/shape) had specific effects on improving WM performance [9]. Specifically, pitch/shape congruency only had an effect when participants focused on the visual modality, whereas pitch/elevation and audiovisual numerosity congruency helped participants when they had to attend to the auditory modality. In addition, the congruency relationship between sounds and visual position had an effect on recall. For example, Marian et al. (2021) found that only congruent environment sounds enhanced memory for where objects were positioned, although auditory stimuli did not offer meaningful spatial information on the objects’ locations [5]. In short, from these studies, the multisensory correspondence of different materials can differentially accelerate memory performance under different sensory modality conditions. However, it is still unknown whether multisensory congruency can slow memory retrieval.

The above studies show that multisensory congruency relationships can affect WM, but the effect of incongruency relationships in multisensory situations is often ignored. Although some studies have revealed that incongruency between targets and irrelevant stimuli can lead to distractions [8,11], researchers also found that incongruent audiovisual stimuli can improve WM. For example, Duarte et al. (2022) found that incongruent sound can speed up the search for simultaneously presented pictures [12]. This result indicated that exposure to meaningful auditory stimuli initiates deeper or more complex semantic processing of vision than the perceptual representation of audiovisual stimuli itself, regardless of whether the stimuli match each other. Marian attributed the advantage of incongruent multisensory stimuli to novelty or the relative sensitivity of the sensory system to incongruent timing [5], but Li et al. (2022) suggested that dissonant sounds may improve subsequent memory performance by increasing alertness [13]. Therefore, these studies demonstrated that incongruent multisensory stimuli can exert significant influence on WM performance and increase individual alertness due to their features.

The effect of multisensory congruency on WM mainly focuses on semantic matching, such as picture and sound (e.g., a picture of a dog and the sound “woof” or a picture of a dog and an oral stimulus of “dog”) [3,14] and written text and sound (e.g., the word “dog” with the sound “woof” or the word “dog” with the spoken word “dog”) [15] and is less focused on matching other stimulus attributes. Even in studies of vocabulary, little consideration has been given to the impact of the audiovisual matching of lexical characteristics. Correspondingly, the characteristically (in)congruent bimodal presentation of words belongs to multisensory correspondence according to Brunetti et al.’s study [7], which refers to whether lexical characteristics presented in both visual and auditory modalities are congruent or incongruent. Concreteness and abstractness are two different characteristics of words [16]. Studies on both cognitive and neural mechanisms have demonstrated that the brain processes concrete and abstract words differently [17,18]. Previous studies have found a concrete effect in which concrete words (e.g., “flowers”) are processed faster and more accurately than abstract words (e.g., “free”) [19], while anti-concrete effects have also been reported [17]. Based on dual coding theory [20], all verbal stimuli initially activate representations in the semantic system, but concrete words can activate information based on mental imagery, while abstract words lack the involvement of the image system. In addition, the context availability model [21] argues that concrete words may have greater relevance to other information than abstract words in semantic memory [22]. For abstract words, mental processes and emotions are required to unify the relevance between them into coherent concepts [23]. Overall, many studies have shown that there are discrepancies in the processing of concrete words and abstract words, but it is unclear whether the simultaneous audiovisual presentation of words with the same or different characteristics affect WM retrieval.

The present study aimed to confirm and further explore the concept that the multisensory congruency relationship can affect WM performance by adopting an n-back paradigm [24]. We investigated the effect of the matching of word characteristics in multisensory stimuli on WM representation for the first time and manipulated attention focus toward the visual or auditory modality. The selective attention manipulation method for the sensory modality during multisensory coding has been widely used in traditional multisensory integration [25] and multisensory recognition memory studies [14]. Due to the fact that both visual and auditory text are processed in the phonological loop [1], we hypothesized that compared with words with incongruent characteristics, words with congruent characteristics in multisensory conditions will cause greater conflict and thus slow unisensory WM retrieval. In addition, because the neural representation of abstract words is more dependent on the verbal system, while the representation of concrete words involves more mental imagery and is more dependent on the perceptual system [26], we hypothesized that the WM retrieval of both auditory and visual presentations of concrete words under visual retrieval conditions would be slower than that under auditory retrieval conditions. In addition, the WM retrieval of both auditory and visual presentations of abstract words under auditory retrieval conditions would be slower than that under visual retrieval conditions.

## 2. Method

### 2.1. Participants

A statistical power analysis in G*Power version 3.1.9.7 Heinrich-Heine-Universität Düsseldorf, Düsseldorf, Germany [27] was performed for sample size estimation. The projected partial η^2^ was determined with reference to a similarly designed three factorial within-subject experiment, and the value was set as 0.25. The two-tailed alpha level was set to 0.05, the power value was set to 0.80, the number of groups was set to 1, and the number of measurements was set to 8. The calculations indicated that a sample size of 30 was needed. Thus, we recruited a total of 30 participants (15 women, 15 men; age range = 18~24 years; mean age = 20.13 years; SD = 1.46) from campus to participate in the experiment. All participants had normal or corrected-to-normal vision and hearing, were right-handed, were reported to not have mental illness, and had not participated in a similar experiment previously. Individuals were compensated the same amount of money for their participation. After receiving a full explanation of the experiment and potential risks, all participants provided written informed consent in accordance with the Code of Ethics of the World Medical Association (Declaration of Helsinki), and the study protocol was approved by the Ethics Committee of Soochow University.

### 2.2. Stimuli and Apparatus

#### 2.2.1. Assessment of the Stimuli Materials

The experimental materials included 24 abstract words and 24 concrete words was shown in Table 1. First, based on the studies conducted by using Chinese words and the “Modern Chinese Dictionary 6th Edition”, subcategories of concrete words and abstract words were selected according to the concreteness and abstractness of words. According to the relevant criteria adopted by predecessors, artificial screening was carried out (mainly including familiarity, concreteness, emotional arousal, and word count), and 37 abstract words and concrete words were selected as alternative stimuli materials. Then, 71 university students who were all aged 18~25 years with normal or corrected-to-normal vision and who had not participated in a similar assessment task were recruited to participate in the assessment of the stimuli materials. The material was assessed by questionnaire survey. Participants were asked to score the familiarity, concreteness, and emotional arousal of 74 words in the alternative stimulus on a scale of 1 to 5 points (1 represents the lowest level, 5 represents the highest level). Combined with the results of the questionnaire, an independent sample t test on the familiarity of abstract words and concrete words was conducted. The results revealed that there was no significant difference in familiarity, *t*(46) = 1.998, *p* > 0.05. Moreover, recent studies have shown that abstract words can elicit more inner properties than concrete words, such as emotions and interoception [28,29]. We conducted an independent sample t test on the emotional arousal of abstract words and concrete words. The results indicated a significant difference in emotional arousal, *t*(46) = 8.1, *p* < 0.001. Finally, an independent sample t test was conducted on the concreteness of the two selected words. The results showed that there was a significant difference in the concreteness of the two types of words, *t*(46) = 26.83, *p* < 0.001. Overall, familiarity, concreteness, and emotional arousal were matched between the concrete and abstract words, as shown in Table 2.

#### 2.2.2. Other Stimuli and Apparatus

All words were presented visually and audibly in a 2-back task [24]. The size of each visual stimulus was 3.06° × 1.72° and was equally distributed across experimental conditions. All auditory stimuli were recorded by the author with the same intonation and emotion and modified with audio-editing software (Adobe Audition version 2020) according to the following parameters: 16 bit and 44,100 Hz digitization, fade in and out, transmitting through both ears at the intensity level of 75 dB. Stimulus presentation, conditions, pseudorandomization, and the recording of responses were all controlled by scripts prepared in Presentation software (Neurobehavioral Systems Inc., Albany, CA, USA; https://www.neurobs.com/ accessed on 5 March 2022.) under the Windows 10 programming environment. Each visual stimulus was presented on a 27-inch ASUS PG 279 display with a screen resolution of 1920 (horizontal) × 1080 (vertical) pixels and a refresh rate of 60 Hz on a black background (RGB value: 0, 0, 0). The central fixation was a white “+” (0.57° × 0.57°); the monitor was located 60 cm away from the participants. Auditory stimuli were presented at a comfortable level through headphones (YINDIAO Q2). The experiment was carried out in a soundproofed, quiet, and dimly lit room where participants used a keyboard (DELL 100) to make key presses, and their responses were recorded.

### 2.3. Design and Procedure

The present experiment consisted of a 2 congruency of audiovisual word characteristics (congruent vs. incongruent) × 2 characteristics of target words (concrete vs. abstract) × 2 unisensory retrieval modalities (visual vs. auditory) within-subject design. The experimental designed a 2-back task based on the study by Soveri et al. [16]. Under the unimodal encoding condition, if the current stimulus was the same as the second stimulus thereafter, it was a target, and if it was different, it was a nontarget. Under the visual retrieval conditions of audiovisual (AV) conditions, if the current visual stimulus was the same as the second subsequent visual stimulus, it was a target, and if it was different, it was a nontarget. Under the auditory retrieval conditions of AV conditions, the target stimulus was similarly presented only in the auditory modality. The possible conditions for the experiment were shown in Figure 1. Pseudorandom sequences were used in the experiment in which we excluded possible 3-back trials and controlled the number of 1-back trials in each block to two to prevent confounding. In addition, under the visual retrieval condition of AV conditions, the auditory stimulus identical to the visual target was not present in the two previous or subsequent trials to prevent confounding when the current trial contained a visual target. Of course, the same was true under the condition of auditory retrieval under AV conditions.

In a gesture to measure the WM of concrete words and abstract words under the unimodal encoding condition and taking it as the baseline, a unimodal encoding condition was also added to the experiment. The 2-back task was administered in 16 blocks, of which the ratio of V:A:AV stimuli was 1:1:1 (2 blocks each for visual concrete words and abstract words, 2 blocks each for auditory visual concrete words and abstract words, and 1 block for each of the 8 AV conditions). The order of the 16 blocks was different for each participant. Each block contained 48 trials, for a total of 768 trials, with a target probability of 37.5%, that is, 18 per block. Because the congruency in the conditions refers to whether the characteristics of the audiovisual presentation of words are congruent rather than whether audiovisual words are the same, the experiment controls congruency under the condition of the audiovisual presentation of congruent word characteristics. The trial ratio of the two types of the same and different audiovisual words was 1:1, and the control of three different types of targets (audiovisual word identical matching, such as the 2-back cue and target are both A:apple and V:apple; audiovisual words are not identical but match, such as the 2-back cue and target are both A:clothes and V:apples; and only target matching, such as the 2-back cue A:clothes, V:apple, target A:banana and V:apple in the concrete visual retrieval condition) was 1:1:1. Moreover, to avoid the error caused by the sequence effect on the experimental results, the sequences in each block were pseudorandomly processed three times, and the blocks were randomized so that each participant was faced with a different sequence during the experiment.

Each block started with the presentation of a fixation cross (500 ms) and continued with a sequence of synchronized visual/auditory/audiovisual stimulus pairs. Each stimulus had a duration of 500 ms (the visual and auditory stimuli were presented simultaneously in the AV condition, and both were presented for 500 ms), followed by a blank interval of 2500 ms (ITI), resulting in an interstimulus interval of 3000 ms (SOA). After the end of each block, participants were required to rest for 30 s, and the next block requirement was displayed on the screen. After the experimenter repeated the following instructions and ensured that the participant understood the requirement, the next block could be started after the rest period. Participants were asked to press a button with the right index finger when the current stimulus was the target and press another button with the right middle finger if it was a nontarget. The participant had to respond by pressing the button as quickly and accurately as possible. The mapping between the two response buttons was counterbalanced between participants. Before the formal experiment, each participant was required to complete practice experiments for each condition, the stimulus duration time was the same as that in the formal experiment, and correct/error feedback followed each trial. The formal experiment did not begin until the participants understood and could accurately repeat the experimental requirements. An experimental procedure example was shown in Figure 2.

## 3. Results

Accurate response rates (ACRs) and reaction times (RTs) were recorded for the sixteen blocks. Accuracy rates were calculated as the percentage of correct responses (correct hits and correct rejections). Trials with no responses or RTs ± 2 SDs beyond the mean RT were not included in the RT analysis [30]. This resulted in the exclusion of 1.1% of trials for the Abs-Test V condition, 1.2% of trials for the Con-Test V condition, 0.8% of trials for the Abs-Test A condition, and 0.9% of trials for the Con-Test A condition in the unisensory condition. Under the AV condition, this resulted in the exclusion of 0.6% of trials for the Cabs-Test A condition, 0.6% of trials for the ICabs-Test A condition, 0.8% of trials for the Cabs-Test V condition, 0.6% of trials for the ICabs-Test V condition, 0.5% of trials for the Ccon-Test A condition, 0.6% of trials for the ICcon-Test A condition, 0.7% of trials for the Ccon-Test V condition, and 0.5% of trials for the ICcon-Test V condition.

**Proportion correct** A 2 congruency of audiovisual word characteristics (congruent and incongruent) × 2 characteristics of target words (concrete and abstract) × 2 unisensory retrieval modalities (V and A) repeated-measures ANOVA was conducted, revealing a significant main effect of unisensory retrieval modalities, *F*(1, 29) = 18.9, *p* < 0.001, $\eta_{p}^{2}$ = 0.40. This result indicated significantly better WM retrieval performance in the auditory modality (95.4%) than in the visual modality (93.3%). There were no significant main effects of congruency of audiovisual word characteristics (*p* = 0.095). In addition, the two-way interaction between the congruency of audiovisual word characteristics and the characteristics of the target word (*p* > 0.05), between the congruency of audiovisual word characteristics and unisensory retrieval modalities (*p* > 0.05), and between the characteristics of the target word and unisensory retrieval modalities were all not significant (*p* > 0.05). Additionally, no three-way interaction was found to be significant (*p* > 0.05).

**Reaction time** The ACRs for visual and auditory WM retrieval performance reached a ceiling in all encoding patterns (above 90%). A 2 congruency of audiovisual word characteristics (congruent and incongruent) × 2 characteristics of target words (concrete and abstract) × 2 unisensory retrieval modalities (V and A) repeated measures analysis of variance (ANOVA) was conducted and revealed a significant main effect of congruency of audiovisual word characteristics, *F*(1, 29) = 10.28, *p* = 0.003, $\eta_{p}^{2}$ = 0.26, indicating a faster retrieval response under the characteristically incongruent condition (668 ms) than under the characteristically congruent condition (709 ms). The results also showed a significant main effect of unisensory retrieval modalities, *F*(1, 29) = 24.29, *p* < 0.001, $\eta_{p}^{2}$ = 0.46, revealing a faster response to the visual retrieval modality (653 ms) than to the auditory retrieval modality (724 ms). Additionally, there was only a significant two-way interaction between the congruency of word characteristics and unisensory retrieval modalities, *F*(1, 29) = 10.04, *p* = 0.004, $\eta_{p}^{2}$ = 0.26. Importantly, the interaction among the three factors was significant, *F*(1, 29) = 5.05, *p* = 0.032, $\eta_{p}^{2}$ = 0.15.

To evaluate the effect of the congruency of the characteristics of words with an audiovisual presentation on subsequent unisensory modality WM retrieval under target words with different characteristics, two separate 2 congruencies of audiovisual word characteristics (congruent and incongruent) × 2 unisensory retrieval modalities (V and A) repeated measures ANOVAs were conducted (Figure 3). For the condition in which the target word characteristics were abstract, significant main effects of congruency of audiovisual word characteristics were observed, *F*(1, 29) = 8.03, *p* = 0.008, $\eta_{p}^{2}$ = 0.22, revealing significantly faster WM retrieval under the incongruent audiovisual word characteristics condition (664 ms) than under the congruent audiovisual word characteristics condition (711 ms). This result also showed a significant main effect of unisensory retrieval modalities, *F*(1, 29) = 10.73, *p* = 0.003, $\eta_{p}^{2}$ = 0.27, indicating significantly faster WM retrieval in visual modality (655 ms) than in the auditory modality (719 ms). There was no significant interaction between the congruency of audiovisual word characteristics and unisensory retrieval modalities (*p* = 0.58).

For the condition in which the target word characteristics were concrete, significant main effects of unisensory retrieval modalities were observed, *F*(1, 29) = 16.84, *p* < 0.001, $\eta_{p}^{2}$ = 0.37, revealing that significantly faster WM retrieval was observed in the visual modality (651 ms) than in the auditory modality (729 ms), but the main effect of congruency of audiovisual word characteristics was marginally significant (*p* = 0.059). Importantly, the interaction between the congruency of audiovisual word characteristics and unisensory retrieval modalities was significant, *F*(1, 29) = 15.31, *p* = 0.001, $\eta_{p}^{2}$ = 0.35. For congruency of audiovisual word characteristics, a post hoc analysis with Bonferroni correction only revealed a significant difference from the incongruent word characteristics under auditory modality and under the visual modality conditions (*p* < 0.001). These results indicated significantly faster WM retrieval in the visual modality (607 ms) than in the auditory modality (738 ms). There was no significant difference between the visual modality and the auditory modality under the congruent word characteristics condition (*p* = 0.31). For unisensory retrieval modalities, a post hoc analysis with Bonferroni correction only revealed a significant difference between congruent word characteristics and incongruent word characteristics under the visual modality condition (*p* = 0.002). This result revealed significantly faster WM retrieval under the incongruent audiovisual word characteristics condition (607 ms) than under the congruent audiovisual word characteristics condition (694 ms). There was no significant difference between congruent word characteristics and incongruent word characteristics under the auditory modality condition (*p* = 0.31).

To evaluate the effect of the characteristics of the target word on subsequent unisensory modality WM retrieval under different congruencies of audiovisual word characteristics, two separate 2 characteristics of the target words (concrete and abstract) × 2 unisensory retrieval modalities (V and A) repeated measures ANOVAs were conducted (Figure 4). For the congruent audiovisual word characteristics conditions, a significant main effect of unisensory retrieval modalities was observed, *F*(1, 29) = 10.76, *p* = 0.003, $\eta_{p}^{2}$ = 0.27, revealing that significantly faster WM retrieval occurred in the visual modality (689 ms) than in the auditory modality (729 ms). However, no significant main effect of the characteristics of the target word was observed (*p* = 0.75), and there was no significant interaction between the characteristics of the target word and unisensory retrieval modalities (*p* = 0.48).

For the incongruent audiovisual word characteristics conditions, a significant main effect of unisensory retrieval modalities was observed, *F*(1, 29) = 23.04, *p* < 0.001, $\eta_{p}^{2}$ = 0.44, revealing significantly faster WM retrieval in the visual modality (618 ms) than in the auditory modality (719 ms). There was no significant main effect of the characteristics of the target word (*p* = 0.58). Notably, the interaction between the characteristics of the target word and unisensory retrieval modalities was significant, *F*(1, 29) = 9.25, *p* = 0.005, $\eta_{p}^{2}$ = 0.24. For the unisensory retrieval modality conditions, a post hoc analysis with Bonferroni correction only revealed a marginally significant difference from the auditory modality under abstract target words and under concrete target words (*p* = 0.058). This result indicated significantly faster WM retrieval for the abstract word conditions (699 ms) than for the concrete word conditions (738 ms). There was no significant difference between abstract words and concrete words under the visual modality retrieval conditions (*p* = 0.24). For the characteristics of target word conditions, a post hoc analysis with Bonferroni correction revealed a significant difference between auditory retrieval and visual retrieval under abstract word conditions (*p* = 0.006 < 0.01), indicating significantly faster WM retrieval in the visual modality (628 ms) than in the auditory modality (699 ms). In addition, another post hoc analysis with Bonferroni correction revealed a significant difference between auditory retrieval and visual retrieval under concrete word conditions (*p* < 0.001), indicating extremely significantly faster WM retrieval in the visual modality (607 ms) than in the auditory modality (738 ms).

## 4. Discussion

The present study aimed to evaluate the effects of characteristically (in)congruent bimodal presentation of words on subsequent unisensory WM retrieval. The presence of lures, along with the fact that all stimuli in the experiment were a perfect bimodal repetition of samples, allowed us to exclude the possibility that participants simply relied on a sense of familiarity to achieve a correct response [31]. According to the reaction time results, this study produced two novel findings. First, for characteristically incongruent multisensory encoding, only under the auditory retrieval condition was the response to abstract words faster than that to concrete words. Notably, regardless of whether the target was concrete or abstract, for characteristically incongruent multisensory stimuli, the speed of WM under the visual retrieval condition was significantly faster than that under the auditory retrieval condition. Furthermore, the advantage of visual WM retrieval under the concrete word condition was far greater than that under the abstract word condition. Second, for the target concrete words accompanied by abstract words presented in another sensory modality, visual WM retrieval was faster than auditory WM retrieval. Moreover, for concrete words in visual WM retrieval, WM retrieval under characteristically incongruent conditions was faster than that under characteristically congruent conditions.

### 4.1. The Multisensory Characteristically (In)congruent Effect

The results elaborately revealed faster verbal WM retrieval under characteristically incongruent multisensory conditions than under characteristically congruent multisensory conditions, which was closely related to visual and auditory verbal WM processing. The result supports the idea that auditory verbal input has direct access to the phonological store, whereas visually presented materials must be converted into a phonological store [32]. According to experimental evidence on the semantic congruent effect on WM [5,33], if visual and auditory words are processed in two different slave systems, WM retrieval under characteristically congruent conditions may not be significantly different or may be even faster than that under characteristically incongruent conditions. We suggest that multisensory characteristically congruent encoding means that two codes are treated similarly with respect to word characteristics in the same slave system, which likely leads to overuse of the cognitive resources that process such characteristics. For multisensory characteristically incongruent encoding, however, due to the different characteristics of words, further processing activates different brain regions and uses different cognitive resources [16,34]. Thus, compared with characteristically incongruent conditions, visual and auditory presentation of words with the same characteristics makes WM retrieval more difficult.

Additionally, the results showed that the ignored modality still influenced memory performance, suggesting that the stimuli to be ignored were nevertheless processed at a semantic level, which is consistent with some studies on multisensory WM with the selective attention method [35]. In particular, Santangelo and Macaluso (2013) revealed that working memory and divided attention utilize a common, limited-capacity pool of processing resources in the overlapping brain region (i.e., the left intraparietal sulcus) [36]. This also explains why words presented in another sensory modality affect WM retrieval even under modality-specific retrieval conditions.

### 4.2. Words with Different Characteristics in Another Sensory Modality Have Differential Effects on WM Retrieval of Concrete Words and Abstract Words

In the present study, we found that under characteristically incongruent conditions, only the auditory WM retrieval of abstract words was faster than that of concrete words. This result indicated that the impact of visual concrete words on the retrieval of auditory abstract words is less than that of visual abstract words on auditory concrete words, which can be comprehended in terms of both verbal working memory and the difference between concrete and abstract words. On the one hand, some neural studies of verbal working memory have also demonstrated that visual letter strings are verbalized as subvocal speech during encoding [37], and participants actively recall and rehearse subvocal speech in their phonological loop during maintenance [38]. Therefore, the task in the experiment can be transformed into a concurrent articulation task in a sense. Additionally, evidence has revealed that concurrent articulation abolishes the phonological similarity effect for visual but not auditory items [39]. However, the maintenance of abstract words, compared to the maintenance of concrete words, is likely more reliant on articulatory rehearsal processes [40]. We tentatively suggest that the greater dependence on phonological processes leads to the ability of auditory abstract words to abolish the phonological similarity effect for visual words better than auditory concrete words, resulting in relatively better WM performance. Crucially, this finding may reflect the important role of top-down control mechanisms in processing abstract concepts: Because abstract words are semantically diverse, a degree of control is necessary to suppress irrelevant information and facilitate retrieval of the appropriate meaning [41]. In addition, according to dual-coding theory [20], this phenomenon occurs precisely because abstract words are more dependent on phonological processing and because abstract words seem to be weakly affected by the representation formed by concrete words. On the other hand, the slower auditory WM retrieval of concrete words under characteristically incongruent conditions may be related to the characteristics of words. Based on dual-coding theory [20], even though auditory concrete words directly enter the phonological loop, they can form visual semantic presentation through the image system. In addition, a recent study revealed that some type of visual image might be evoked, at least to some extent, even by abstract and poorly imageable nouns [42]. Therefore, under characteristically incongruent conditions, when encoding, the visual representation generated by auditory concrete words may compete with the relevant visual representation of visual abstract words, resulting in slower WM retrieval of auditory concrete words.

Moreover, this result supports the idea that selective attention can leave more resources for subsequent higher-order processing and promote the generation of coherent multisensory representations [3]. We hypothesize that the encoding of concrete words with selective attention seems to be too active in utilizing visual representation to form coherent multisensory representation. Therefore, the representation of other sensory modalities may be incorrectly combined into multisensory encoding, by which the interference caused may lead to slower WM retrieval. Additionally, our results seem to support anti-concrete effects [18] rather than the concreteness effect [19]. However, to some degree, compared with abstract words, concrete words can indeed achieve an advantage by additional activation of the image system or by having more contextual information [43]. Nevertheless, such an advantage seems to be task specific. Some evidence also shows that once imageability and context availability are controlled, there is a residual advantage for abstract word processing [44]. Similarly, in the current study, we suggest that visual abstract words may prevent the imageability and context availability of the auditory system, resulting in faster WM retrieval under abstract word conditions than under concrete conditions.

Furthermore, the results of the study showed that regardless of whether the characteristics of the target word were concrete or abstract, under the characteristically incongruent condition, visual WM retrieval was faster than auditory WM retrieval. On the one hand, evidence has shown that with concrete targets, there was interference only for words that were semantically similar [45]. For concrete words, under the visual retrieval condition, the auditory abstract word was not semantically similar to visual concrete words and thus generated less interference with the encoding of concrete words. In addition, auditory abstract words directly enter the phonological loop [32], and their process is more reliant on articulatory rehearsal processes [40], which can hardly generate any visual representation to compete with visual concrete words.. Thus, in general, auditory abstract words have little impact on visual WM retrieval of concrete words. However, for concrete words under auditory retrieval conditions, as mentioned above, the representation generated by visual abstract words interferes with the encoding of auditory concrete words, resulting in difficult auditory WM retrieval. On the other hand, for abstract words, because both visual and auditory words are processed in the phonological loop [37], as explained above, the experimental task can be transformed into a concurrent articulation task in a sense. Such concurrent articulation abolishes the phonological similarity effect for visual but not auditory items [39]. Additionally, it should be noted that evidence has shown that, with abstract targets, there was interference only for associated words but not for words that were semantically similar [41]. Thus, we suggest that the presentation of concrete words in auditory or visual modality does not interfere with the encoding of abstract words under a task similar to concurrent articulation tasks. Alternatively, regarding the sensory modality, this result is consistent with previous multisensory WM studies that used semantically incongruent or cross-modal unmatched audiovisual materials [3,7,33], suggesting a dominant role of the visual modality in multisensory representation [46]. Overall, slower auditory memory retrieval may be caused by slower perceptual processing [3], and the treatment or maintenance of verbal text uses more phonological loop resources [1]. Thus, similar to other stimuli, transforming an auditory verbal stimulus into a perceptual representation may be more costly with respect to time than transforming a visual verbal stimulus into a perceptual representation [47].

Additionally, interestingly, the advantage of visual WM retrieval under the concrete word condition was far greater than that under the abstract word condition. This result once again proves that concrete words seem to invoke the image system and verbal system to promote the generation of robust representations, leading to a stronger visual advantage compared with abstract words [20]. Additionally, according to the context-availability model [21], Xiao et al. (2016) found an increased P600-like component in response to concrete words relative to that in response to abstract words, likely reflecting the retrieval of more contextual details [48]. Other evidence shows that concrete concepts are more easily visualized than abstract concepts; the former are connected with highly visually specific items, whereas abstract concepts may only be visualizable through the use of loosely connected symbols [49]. Thus, compared with abstract words, concrete words can activate more contextual information when encoding, which should facilitate WM retrieval. Alternatively, a recent study revealed that because of their complexity, abstract words might lead to higher uncertainty, less confidence in their meaning, and stronger involvement of metacognition and inner speech than concrete words [50]. Therefore, due to the complexity of abstract words, when performing visual encoding, abstract words tend to use more cognitive resources than concrete words, which may elicit a relatively slower response during WM retrieval.

### 4.3. Concrete Words and Abstract Words Presented in the Other Sensory Modality Have Different Effects on WM Retrieval of Concrete Words

In the present study, for the concrete words, the characteristically congruent effect was observed only under the visual retrieval condition. This result indicated that auditory concrete words have a stronger effect on visual WM retrieval of concrete words than auditory abstract words. According to Baddeley’s opinion, auditory verbal information enters directly into the phonological loop [32]. We suggest that such a result should be relevant to the characteristics of words. Firstly, according to dual-coding theory, concrete words can activate both semantic and image-based systems, while abstract words are primarily or exclusively represented in the verbal system [20]. Many recent relevant neurological studies have also found that abstract concepts rely more on the language system and that concrete concepts rely more on sensory-motor systems [49]. Therefore, the interference with visual representation of concrete words resulting from an auditory abstract word occurs largely on a phonological rather than semantic level. Additionally, visual representation has a dominant role in multisensory encoding, so phonological interference is unlikely to obviously affect WM retrieval. Additionally, the present study may support the idea that auditory concrete words can also generate visual representation, of which the conflicting representation would interfere with concrete words and thus result in slow visual WM retrieval.

Second, according to the context availability model, we tentatively suggest that auditory concrete words provide more contextual information from semantic memory compared with auditory abstract words [21]. The large amount of information provided by visual and auditory stimuli at the same time may lead to encoding difficulties, which could affect the visual WM retrieval of concrete words. An influential theory put forward by Warrington [51] suggests that abstract concepts are organized by semantic association and concrete ones are organized by semantic similarity, with concepts belonging to the same semantic category. Importantly, a previous study on the organization of abstract and concrete concepts showed that significant interference effects were observed for concrete items that were blocked by semantic similarity (category) but not when they were blocked by association [23]. Similarly, in the present experiment, auditory concrete words were more likely to be organized together with visual concrete words during WM encoding because of semantic similarity, while auditory abstract words were not. Visual and auditory representations organized together were inherently difficult for WM encoding, which may reduce the stability of visual concrete word representations and thus result in more time spent on WM retrieval. However, because auditory abstract words could not be organized with visual concrete words, the representation of visual concrete words was slightly affected, and visual WM retrieval was faster than that under the characteristically congruent condition. Furthermore, studies show that most concrete concepts can be readily organized into categories based on the perceptual and functional characteristics that overlap within a category [26]. Abstract concepts, however, are more difficult to describe based on their semantic neighbors and features, leading to the assumption that abstract concepts are semantically “impoverished” [52]. Therefore, we tentatively suggest that the characteristic overlapping of concrete words under the characteristically congruent condition means more competition for the formation of the visual representation because concepts in a neighborhood tend to be densely interconnected and activate each other [22]. However, for auditory abstract words, the competition between visual and auditory representation is small compared with the characteristically congruent condition to some degree, so the impact on WM retrieval is less.

Contrary to the result for concrete words under the visual retrieval condition, the characteristically congruent effect was not found under the auditory retrieval condition. The results indicated that, regardless of whether the words were concrete or abstract, the superior robust representation for material presented auditorily relative to that presented visually supports the idea that verbal materials presented auditorily have direct access to the phonological store, whereas visual information must be recoded via articulation [32]. Some studies argue that the modality effect is actually a learning effect [39]. Specifically, adults, even very literate adults, have substantially more experience in mapping from acoustics to meaning or acoustics to articulation than they do in mapping from orthography to meaning or acoustics [39]. In addition, there has been limited evidence that auditory verbal working memory contributes to understanding speech in noise [53]. Our results seemed to demonstrate that auditory verbal working memory is more resistant to interference. Additionally, such a result indicates that, for concrete words under auditory retrieval conditions, the visual representation generated by visual abstract words may be similar to visual concrete words, which are mainly based on the word form [54]. Thus, the competitions with visual representation generated by auditory concrete words are likely to be the same.

Moreover, only when abstract words were presented in the other sensory modality was visual retrieval faster than auditory retrieval. The dominant role of visual information on WM in multisensory studies seems to be a relatively stable phenomenon [3,33]. Studies have shown that directional information flow occurs from the auditory cortex to the hippocampus during encoding [37], and concrete concepts rely more on sensory-motor information [26,49]. Thus, higher costs may result in more time being spent on auditory WM retrieval than on visual WM retrieval in multisensory conditions [55]. Alternatively, as mentioned above, visual abstract words can also generate certain visual representations [42,54], which may interfere with the encoding of concrete words under auditory retrieval conditions. Nevertheless, auditory abstract words directly enter the phonological loop [32], and rely more on the language system [26,49]. Thus, abstract words have less interference on visual concrete word encoding under visual retrieval. Therefore, the increase in the absolute value of the gap between the RTs of the auditory retrieval condition and visual retrieval condition leads to the existence of a significant visual dominance effect. Additionally, multisensory studies on WM reported that when there was an incongruent match or noncorrelation between visual and auditory stimuli, visual retrieval was still observed to be faster than auditory retrieval [3,7,15]. In the present study, concrete words and abstract words were processed and retained in different ways and activated different brain cortices [16,32]. Therefore, for concrete words under characteristically incongruent conditions, the abstract words presented in another sensory modality were similar to incongruent or irrelevant audiovisual stimuli in previous studies [3], which produced similar results.

However, notably, for concrete words under the characteristically congruent condition, there was no significant difference between the visual retrieval condition and auditory retrieval condition. The results indicated that the representation of auditory concrete words has a greater impact on the WM retrieval of visual concrete words, which influences the dominant role of visual stimuli in multisensory encoding. Recent evidence suggests that concrete words are firmly grounded in the sensorimotor system [28,49], including the perceptual and mental image generation systems [16]. Thus, we suggest that auditory concrete words can more strongly invoke the image system, through which visual representation is generated. Therefore, the visual representation generated by auditory concrete words may greatly disturb the encoding of visual concrete words. Visual representations of the target may become relatively weak because of another conflicting representation, thus leading to visual stimuli losing their dominant position in multisensory WM retrieval.

## 5. Conclusions

In summary, the present study investigated the impact of characteristically (in)congruent bimodal presentation of words on WM retrieval. The results showed robust differences between the characteristically congruent condition and incongruent condition; that is, for the characteristically incongruent condition under the auditory retrieval condition, the response to abstract words was faster than that to concrete words. Moreover, for concrete words under the visual retrieval condition, WM retrieval was faster in the characteristically incongruent condition than in the characteristically congruent condition. The present findings provided evidence that concrete words tend to be mistakenly encoded with other visual representations in multisensory conditions, thus slowing WM retrieval. In contrast, the encoding of abstract words seemed to better suppress interference, and, accordingly, WM performance for abstract words was better than that for concrete words.

## Figures and Tables

**Figure 1 brainsci-13-00811-f001:**
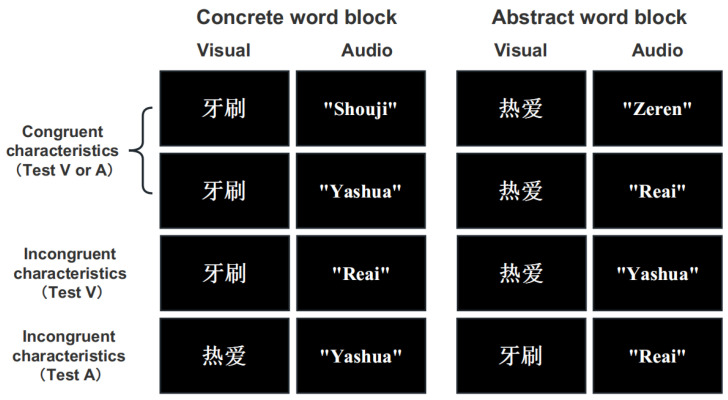
Eight conditions of (in)congruent characteristics in the concrete word block and abstract word block under modality-specific selective attention. Eight conditions separately evaluating the bimodal presentation (i.e., congruent characteristics and incongruent characteristics) for subsequent unimodal retrieval (i.e., V and A) under modality-specific selective attention conditions. Congruency in the experiment refers to whether the characteristics of the words presented as audiovisual stimuli were congruent. The English meanings of the example Chinese words in the figure are as follows: “牙刷”/“Yashua” means toothbrush, “热爱”/“Reai” means enthusiasm, “手机”/“Shouji” means mobile phone, and “责任”/“Zeren” means responsibility.

**Figure 2 brainsci-13-00811-f002:**
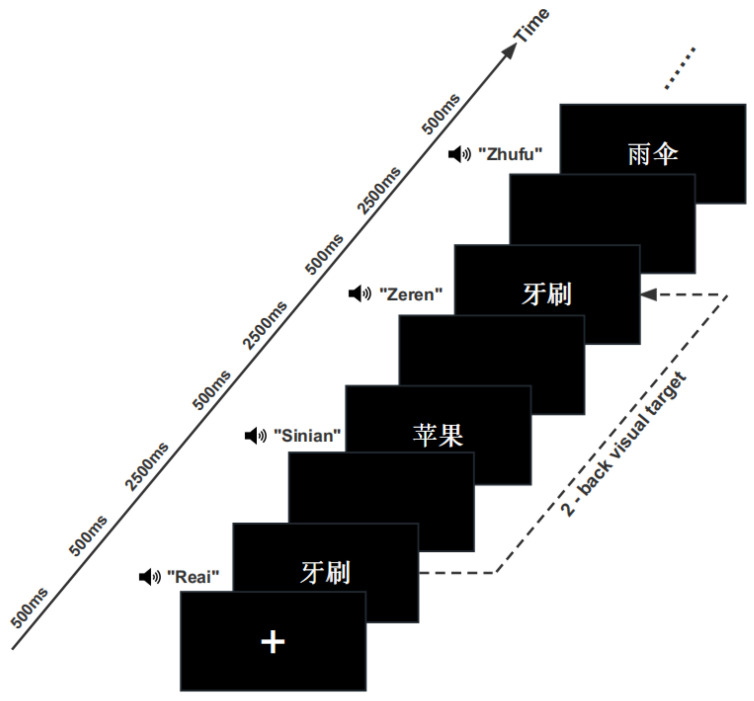
An experimental procedure example of incongruent characteristics of audiovisual presentation of a concrete word block under visual retrieval conditions. A fixation cross was shown for 500 ms, and a stimulus (characteristically congruent with the incongruent audiovisual stimulus) was then presented for 500 ms. A blank screen was shown after a 2500 ms delay and continued with a sequence of stimulus and blank screen presentations. The participants were asked to determine whether the current stimulus was the same as the second stimulus thereafter under the required attentional modality. The English meanings of the example Chinese words in the figure are as follows: “牙刷”/“Yashua” means toothbrush, “苹果”/“Pingguo” means apple, “雨伞”/“Yusan” means umbrella, “热爱”/“Reai” means enthusiasm, “思念”/“Sinian” means yearning, “责任”/“Zeren” means responsibility, “祝福”/“Zhufu” means blessing.

**Figure 3 brainsci-13-00811-f003:**
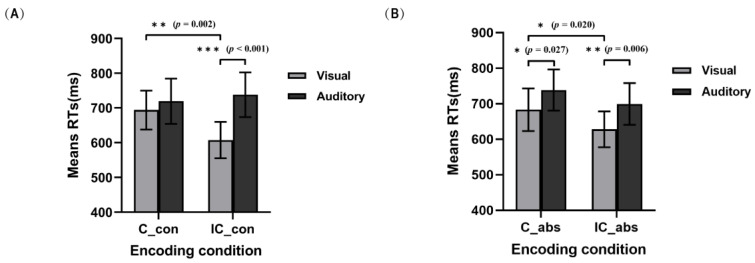
Mean RTs modulated characteristically congruent multisensory effects of unisensory WM retrieval grouped by concrete words condition (**A**) and abstract words condition (**B**). C_con means that there were both visual and auditory presentations of concrete words; IC_con means that only concrete words were presented in the retrieval modality, and abstract words were presented in the other modality; C_abs means that there were both visual and auditory presentations of abstract words; IC_abs means that only abstract words were presented in the retrieval modality, and concrete words were presented in the other modality. Sample size is 30 people. Error bars denote the SE. * *p* < 0.05, ** *p* < 0.01, *** *p* < 0.001.

**Figure 4 brainsci-13-00811-f004:**
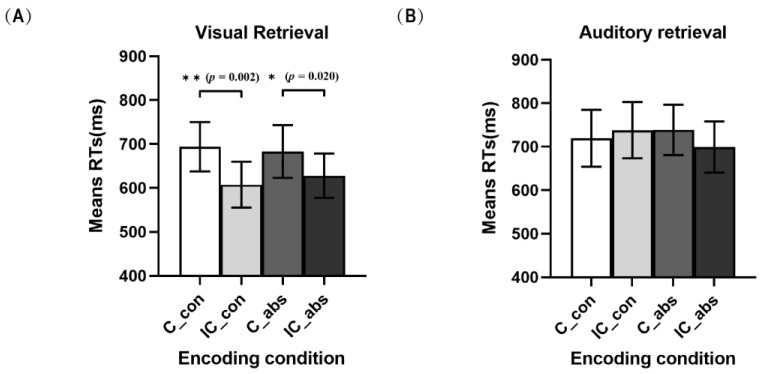
Mean RTs modulated characteristically congruent multisensory effects of unisensory WM retrieval grouped by the visual retrieval condition (**A**) and auditory retrieval condition (**B**). Sample size is 30 people. Error bars denote the SE. * *p* < 0.05, ** *p* < 0.01.

**Table 1 brainsci-13-00811-t001:** Stimulating materials applied in the experiment: 24 concrete words and 24 abstract words.

Set	Concrete Words		Abstract Words	
1	牙刷	toothbrush	自由	freedom
2	书本	book	公平	fairness
3	剪刀	scissor	原理	principle
4	衣服	clothes	思念	yearning
5	眼镜	glasses	热爱	enthusiasm
6	电脑	computer	鼓励	encouragement
7	雨伞	umbrella	精神	spirit
8	苹果	apple	名声	reputation
9	花朵	flower	祝福	blessing
10	香蕉	banana	隐私	privacy
11	书桌	desk	成功	success
12	时钟	clock	希望	hope
13	手机	mobile phone	责任	responsibility
14	西瓜	watermelon	记忆	memory
15	杯子	glass	幸福	happiness
16	篮球	basketball	坚强	fortitude
17	猫咪	cat	魅力	charm
18	电灯	electric light	梦想	dream
19	毛巾	towel	民主	democracy
20	浴缸	bathtub	和谐	harmony
21	石头	stone	友善	kindness
22	哑铃	dumbbell	文明	civilization
23	鞋子	shoes	正直	integrity
24	书包	schoolbag	敬业	dedication

**Table 2 brainsci-13-00811-t002:** The familiarity, concreteness and emotional arousal of 24 concrete words and 24 abstract words were scored in the questionnaire.

Characteristic	Familiarity (Means ± SDs)	Concreteness (Means ± SDs)	Emotional Arousal (Means ± SDs)
Concrete	3.73 ± 0.21	4.13 ± 0.10	3.41 ± 0.24
Abstract	3.82 ± 0.10	3.34 ± 0.10	3.71 ± 0.19

## Data Availability

The data is available from the corresponding author upon reasonable request.

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
