# Peer review of "Semantic Bimodal Presentation Differentially Slows Working Memory Retrieval"

_brainsci, 2023, doi:10.3390/brainsci13050811_

Round 1

Reviewer 1 Report

This study investigated how multisensory congruency affects working memory (WM) retrieval of concrete and abstract words. Results showed that when retrieving visual concrete words, WM retrieval was faster in the characteristically incongruent condition than in the characteristically congruent condition. In contrast, responses to auditory abstract words were faster than those to auditory concrete words in the characteristically incongruent condition. These findings suggest that concrete words in multisensory conditions may interfere with WM retrieval, while abstract words may suppress interference better and lead to better WM performance. It can be considered for publication with minor revisions.

1. In lines 19-20, it was mentioned that a total of 30 participants were recruited, while in lines 39-40, it was stated that 71 university students were recruited for the assessment. Please elaborate.

2. If post hoc methods were used for the one-way ANOVA analysis, it is recommended to specify which method was used in the Methods section.

3. It recommends indicating the exact p-value in the figures 3 and 4, as well as the sample size in the figure captions.

4. Please format references.

Reviewer 2 Report

The paper can be interesting to a specific audience, and has definitely some merits. 

However, it is surely not up to academic standards. Both English language and written style need to be improved drastically and to be revised with the help of a native speaker. The awkwardness of the language and, often, its non-grammatical / non-orthographic nature, appear already from the title (and that is 'epic') and the Abstract, strengthened by a series of typos. Truly, that needs to be fixed, guys. The same is true for the style, with some super-long sentences which are too complicate to be read, and with a plethora of repetitions and redundancies, even at the lexical level, which need to be fixed. Please, correct. 

The Introduction is comprehensive enough, but the Authors should stress more on the intrinsic relevance of their paper in their field of studies and highlight very clearly their research goals and how they plan to achieve them. 

 Where is the Literature Review? 'Scattered' here and there, all over the paper? Why not to develop a proper section, originally entitled "Literature Review", after the Introduction and before the section on the methodology, with a precise list and analysis of the works used and cited in the article and of significant works from the field in general, to make the paper more 'user-friendly' even to a non-specialized audience? 

The Methodology is well explained - despite the related statement by the Authors, I am not sure that the sample, at the numerical level, is indicative enough, but that is a matter of interpretation and preference. 

Should the "Result" section be entitled "Results", or do the Authors mean that we have a 'result in itself'? In this section, the language becomes even 'harsher' than in the other sections, and the results themselves, despite quite understandable, should be double-checked by a Reviewer with more expertise than the one who is writing now in the specific field. 

The Discussion is comprehensive enough, a little redundant, somewhere, but I truly would never dare to 'cut' a Discussion, since, generally, papers are in need of more discussions which are never produced. Is there a Conclusion, in the article? Apparently not. Where is it? Did the Authors forget it in their keyboards? A proper academic paper needs a proper Conclusion, which should be added by the Authors and which should summarize again, like in a 'mirror' with the Introduction, the importance of their research in the current panorama of studies and how they have achieved their epistemological goals. 

All in all, the paper requires truly a lot (a lot, a lot, a lot, a lot) of work, before being considered for publication - despite this, if thoroughly revised, it can be an interesting piece of research. 

Thank you very much. 

As told, the paper is surely not up to academic standards. Both English language and written style need to be improved drastically and to be revised with the help of a native speaker. The awkwardness of the language and, often, its non-grammatical / non-orthographic nature, appear already from the title (and that is 'epic') and the Abstract, strengthened by a series of typos. Truly, that needs to be fixed. The same is true for the style, with some super-long sentences which are too complicate to be read, and with a plethora of repetitions and redundancies, even at the lexical level, which need to be fixed. Please, correct. 

Round 2

Reviewer 2 Report

A lot of work has been invested in trying to improve the paper. 

The article itself still shows some 'weak' spots, but it is reasonably understandable, now, and quite effective. 

It can, therefore, be considered for publication. 

The Authors' effort is appreciated. 

Thank you very much. 

The English language is better, now - despite this, it is not 'native-like' yet, and we should consider to what extent is up to academic standards; possibly, a further round of revision by MDPI specialists can help.